# Surveillance of WHO Priority Gram-Negative Pathogenic Bacteria in Effluents from Two Seafood Processing Facilities in Tema, Ghana, 2021 and 2022: A Descriptive Study

**DOI:** 10.3390/ijerph191710823

**Published:** 2022-08-30

**Authors:** Meldon Ansah-koi Agyarkwa, Godfred Saviour Kudjo Azaglo, Henry Kwabena Kokofu, Ebenezer Kwabena Appah-Sampong, Esi Nana Nerquaye-Tetteh, Emmanuel Appoh, Jewel Kudjawu, Ebenezer Worlanyo, Mariam Fuowie Batong, Amos Akumwena, Appiah-Korang Labi, Mary-Magdalene Osei, Srinath Satyanarayana, Robert Fraser Terry, Marcel Manzi, Japheth A. Opintan

**Affiliations:** 1Environmental Protection Agency, Ministries, Accra P.O. Box MB 326, Ghana; 2Department of Medical Microbiology, University of Ghana Medical School, Accra P.O. Box GP 4236, Ghana; 3World Health Organization Country Office, Accra P.O. Box MB 142, Ghana; 4Center for Operational Research, International Union against TB and Lung Disease, New Delhi 110016, India; 5UNICEF, UNDP, World Bank, WHO, Special Programme for Research and Training in Tropical Diseases (TDR), World Health Organization, 1211 Geneva, Switzerland; 6Department of Medical OCB, MSF-Belgium Headquarters, Rue de Bomel 65, 5000 Namur, Belgium

**Keywords:** SORT IT, One Health, operational research, antimicrobial resistance, surveillance, WHO priority pathogens, effluent, seafood processing facilities, Environmental Protection Agency-Ghana

## Abstract

Antimicrobial resistant (AMR) bacteria in effluents from seafood processing facilities can contribute to the spread of AMR in the natural environment. In this study conducted in Tema, Ghana, a total of 38 effluent samples from two seafood processing facilities were collected during 2021 and 2022, as part of a pilot surveillance project to ascertain the bacterial load, bacterial species and their resistance to 15 antibiotics belonging to the WHO AWaRe group of antibiotics. The bacterial load in the effluent samples ranged from 13–1800 most probable number (MPN)/100 mL. We identified the following bacterial species: *E. coli* in 31 (82%) samples, *K. pneumoniae* in 15 (39%) samples, *Proteus* spp. in 6 (16%) samples, *P. aeruginosa* in 2 (5%) samples and *A. baumannii* in 2 (5%) samples. The highest levels of antibiotic resistance (100%) were recorded for ampicillin and cefuroxime among *Enterobacteriaceae*. The WHO priority pathogens—*E. coli (*resistant to cefotaxime, ceftazidime and carbapenem) and *K.*
*pneumoniae (*resistant to ceftriaxone)—were found in 5 (13%) effluent samples. These findings highlight the need for enhanced surveillance to identify the source of AMR and multi-drug resistant bacteria and an adoption of best practices to eliminate these bacteria in the ecosystem of the seafood processing facilities.

## 1. Introduction

Antimicrobial resistance (AMR) is regarded as one of the greatest global health risks to humans and animals. AMR undermines the ability of antibiotics to effectively prevent and treat bacterial infections, resulting in severe health and economic consequences. It is estimated that about 10 million lives and USD 100 trillion will be lost by 2050 because of AMR, if no high priority actions are taken. AMR is an emerging global issue, but the burden is disproportionately excessive in low-income and middle-income countries [1,2,3].

Whilst the spread of AMR has been attributed to the irrational use of antimicrobials in humans and animals, it is increasingly being recognized that the environment plays a very important role in the development and transmission of AMR, thus, the One Health approach is key to understanding and tackling AMR [4]. The One Health approach recognizes the intrinsic interlinkages between humans, animals and the environment and therefore advocates the need for a collaborative, multisectoral approach in achieving optimal human and animal health outcomes [4]. Surveillance is an essential component of the One Health approach, and to facilitate AMR surveillance the World Health Organization (WHO) has identified and published a list of antibiotic-resistant “priority pathogens”—a catalogue of 12 families of bacteria that pose the greatest threat to human health [5].

AMR bacteria can reach the environment from different sources, such as through untreated effluents discharged from healthcare settings, slaughter houses and industries, and they can contaminate surface and groundwater that serves as potable or irrigation water [6]. The effective management of industrial effluent discharges and the conservation of the natural environment to enhance its sustainability is paramount to achieving the Sustainable Development Goal (SDG 6.3) [7]. This goal has a target of improving water quality by reducing pollution and the contamination of water bodies from industrial and other sources [7].

The increased use of antimicrobials in aquaculture and seafood processing facilities in some geographic areas has led to an increase in AMR in effluent discharges from these facilities. This phenomenon makes seafood processing facilities a potential source of AMR bacteria [6]. Seafood processing facilities use large volumes of water during the processing of fish and the cleaning of equipment [8]. Effluents from seafood processing facilities are rich in organic matter and can serve as an important growth medium for bacteria, resulting in an amplification of antibiotic resistance [9]. Studies have shown evidence that effluent from seafood processing facilities contains multi-drug resistant *Escherichia coli* [6,10].

In Ghana, the Tema fishing harbor enclave undertakes major fishing activities in the Greater Accra region. This enclave houses large seafood processing facilities. Effluent discharges from their production activities are channeled into open communal drains and the Chemu Lagoon, which flows into the sea. This practice has the potential of contaminating the waterbodies and contributing to the burden of AMR bacteria in the local environment.

The Ghana Environmental Protection Agency (EPA) of the Ministry of Environment Science, Technology and Innovation was established by the EPA Act 490 (1994), and it is mandated to monitor environmental pollutants and their impact on the recipients [11]. In line with the One Health approach, there is need for information about the presence of AMR bacteria, especially those that have been highlighted by the WHO as “priority pathogens” in the ecosystem of seafood processing facilities. This is essential for the development of guidelines for routine surveillance and to assess the potential implications for public health. There have not been any studies/reports in the country on the bacterial profile of the effluents from the seafood processing facilities.

In this context, an operational research study was undertaken to: (a) describe the bacterial load and the antibiotic resistance patterns of the bacterial species isolated from the effluents of the two seafood processing facilities; and (b) to assess if WHO priority Gram-negative pathogenic bacterial species were present in effluents discharged from the two seafood processing facilities. The WHO priority pathogens that were of interest per the scope of the study were *Enterobacteriaceae*-carbapenem-resistant, *Enterobacteriaceae*-3rd generation cephalosporin-resistant, *Pseudomonas aeruginosa*-carbapenem-resistant and *Acinetobacter baumannii*-carbapenem resistant.

## 2. Materials and Methods

### 2.1. Study Design

This was a cross-sectional study using secondary data from a pilot surveillance project that was implemented by the EPA for assessing the bacterial profile in effluents of two seafood processing facilities.

### 2.2. Study Setting

#### 2.2.1. General Setting

Ghana is a country located along the Gulf of Guinea and the Atlantic Ocean, in the subregion of West Africa. The country has a population of about 30.8 million people and Greater Accra is the most populous region [12].

#### 2.2.2. Specific Setting

Tema is a city on the Atlantic Coast of Ghana. It is located 25 km east of the capital city, Accra. The largest seaport in Ghana is situated in Tema. There are seafood processing facilities in Tema that produce canned seafood products (tuna). Their source of water for industrial processes is mainly from the Ghana Water Company Limited (GWCL). Anecdotal evidence indicates that their production processes do not involve the use of antibiotics. The effluent discharge from their production activities is channeled into open communal drains and a nearby lagoon (Chemu Lagoon), which flows into the sea. Annual environmental quality reports from the seafood processing facilities, submitted to the EPA, indicate the presence of effluent treatment systems to prevent or reduce bacterial discharge into the surrounding environment. Their treatment procedure consists of an aerobic effluent treatment system that involves the use of poly aluminum chloride and/or polymer to reduce the bacteria load.

### 2.3. Effluent Sample Collection and Laboratory Analysis

The EPA, as part of the surveillance system, collected a total of 38 weekly effluent samples (19 from each seafood processing facility) in 2021 and 2022, as shown in Table 1. Sample collection and analysis were completed using the most probable number (MPN) method, according to the procedures outlined in the Standard Methods for the Examination of Water and Wastewater, 2012 [13]. Effluent grab samples were collected at the final effluent discharge points of each of the two seafood processing facilities. Grab samples are a small representative subset of a larger quantity, taken at a specific time.

The effluent samples were systematically collected into sterile 1 L high density polythene (HDP) bottles, using an aseptic technique, during the morning hours, from each of the two effluent sampling points. We measured the pH and temperature of all samples in situ. The values recorded for pH and temperature are shown in the Appendix A. Thereafter, all samples were transported on ice, within an hour, to the University of Ghana Medical School, Department of Medical Microbiology Laboratory for immediate analysis. Samples were analyzed using a standard multiple tube method using lauryl tryptose broth [14]. Inoculated tubes were incubated at 35 ± 0.5 °C for 24 h; each tube was examined, and the presence or absence of heavy growth, gas and acid reaction was recorded. Incubation was extended for 48 h if no gas or acidic growth was seen. Bacteria load was estimated using the most probable number (MPN) index and recorded as MPN/100 mL. Subsequently, the streaking method was used to isolate bacteria from positive tubes on MacConkey agar (OXOID, UK) for coliforms, and Eosin methylene blue agar (OXOID, UK) for other Gram-negative bacteria. Streaked plates were then incubated for 15–24 h at 37 °C. Bacteria were identified using the matrix-assisted laser desorption ionization time-of-flight (MALDI-TOFF) method [15].

Antibiotic susceptibility testing was performed using the Kirby–Bauer disk diffusion technique according to Clinical Laboratory Standards Institute (CLSI) guidelines [16,17]. A total of fifteen commonly used antibiotics in Ghana were screened. They included tetracycline 30 µg, sulfamethoxazole/trimethoprim 25 µg, gentamicin 10 µg, amikacin 30 µg, ampicillin 10 µg (antibiotics belonging to the ‘Access’ group); cefuroxime 30 µg, cefotaxime 30 µg, ciprofloxacin 5 µg, levofloxacin 5 µg, ceftazidime 30 µg, piperacillin/tazobactam 100/10 µg, cefepime 30 µg, ceftriaxone 30 µg and meropenem 10 µg (antibiotics belonging to the ‘Watch’ group); and colistin 10 µg (belonging to the ‘Reserve’ group) of the WHO AWaRe antibiotic classification system [18].

### 2.4. Laboratory Quality Assurance Procedures

All media were prepared according to manufacturer’s instructions. Media and antibiotics used were quality controlled using the reference organism *E. coli* ATCC 25922, following 2021 CLSI guidelines.

### 2.5. Data Collection, Source of Data and Validation

Data were collected for our operational research study from the surveillance database and included the following: effluent sample identifier and date; bacterial counts in MPN per 100 mL of the effluent sample; bacterial species isolated; and the results of the antibiotic susceptibility testing. All these data were primarily documented in a laboratory notebook and thereafter entered into an electronic database (Microsoft Excel^®^) kept at the Department of Medical Microbiology laboratory, University of Ghana Medical School. This electronic database was shared and maintained in the EPA Environmental Quality Laboratory database. Prior to using the data in the electronic database for our study, we validated it with the raw data contained in the laboratory notebook.

### 2.6. Statistical Analysis

All data were analyzed using RStudio (version 5501.9.2.0, “Ghost Orchid” developed by RStudio PBC, Boston, MA, USA, info@rstudio.com) statistical software. We described the bacterial load (MPN/100 mL) of the effluent samples collected at various timepoints, and the bacterial species that were isolated. Thereafter, we described the isolation of the following species in various samples that were of importance from a surveillance point of view: (a) *Enterobacteriaceae* (*E. coli*, *Klebsiella* spp., *Proteus* spp., *Shigella* spp. and *Salmonella* spp.); (b) *Pseudomonas aeruginosa*; (c) *Acinetobacter baumannii*. Following this, we described the antibiotic susceptibility pattern of these surveillance bacterial species to the 15 antibiotics, which included ascertainment of resistance to certain antibiotics (carbapenems and 3rd generation cephalosporins) that qualifies these bacteria as WHO priority pathogens. The Fisher’s exact test was used to assess the proportions of bacterial isolates showing resistance in effluent samples. We also calculated and presented the multiple antibiotic resistance (MAR) index for each of the bacterial isolates under surveillance by using the formula MAR = a/b, where ‘a’ represents the number of antibiotics to which the test isolate depicted resistance and ‘b’ represents the total number of antibiotics to which the bacterial isolate was evaluated for antibiotic susceptibility [19].

## 3. Results

### 3.1. Bacterial Load in the Effluents

The bacterial load of the 38 samples disaggregated by the source of the effluent samples and the month of collection is given in Figure 1. The bacterial load of the effluent samples from seafood processing facility 1 (SPF-1) was higher than that of seafood processing facility 2 (SPF-2) in most of the samples. On four occasions in 2021, the samples of SPF-1 had the highest measurable bacterial load (1800 MPN/100 mL).

### 3.2. Bacterial Species and Their Antimicrobial Resistance Pattern

The bacterial species isolated from these effluent samples are given in Table 2. Most of the effluent samples contained a mix of bacterial species. The most common species identified included *E. coli*, *M. morganii*, *Pseudomonas* spp., *Klebsiella* spp., *P. mirabillis* and *C. freudii*.

Other bacterial species, such as *Salmonella* spp., and *Shigella* spp. organisms, that were also of significance under surveillance, were not isolated from these effluent samples. The effluent samples in which the bacteria of surveillance significance were isolated are given in Figure 2. In almost all samples *E. coli* was identified. *K. pneumonia*e was isolated predominantly in the samples from seafood processing facility 2 (SPF-2) and in the initial samples collected. *P. mirabillis* and *P. aeruginosa* were isolated from the effluent samples of seafood processing facility 1 (SPF-1) and *A. baumannii* in seafood processing facility 2 (SPF-2).

The antibiotic resistance patterns of the organisms under surveillance are given in Table 3. There was a progressive decrease in antibiotic resistance from the ‘Access’ group through to the ‘Watch’ group. In samples from SPF-1, the highest levels of antibiotic resistance were for ampicillin (63% in *E. coli* and 80% in *K. pneumoniae*) followed by cefuroxime (75% in *E. coli* and 80% in *K. pneumoniae*). In addition, for species isolated from effluents of SPF-2, the highest levels of antibiotic resistance were for ampicillin (87% in *E. coli* and 67% in *K. pneumoniae*) followed by cefuroxime (100% in *E. coli* and 67% in *K. pneumoniae*).

### 3.3. Presence of WHO Priority Pathogens and Multi-Drug Resistance in the Effluents of Seafood Processing Facilities

As seen in Table 3, the following WHO priority pathogens were isolated: *E. coli* resistant to cefotaxime was isolated from eight effluent samples; *E. coli* resistant to ceftazidime was isolated from one effluent sample; *E. coli* resistant to meropenem and *K. pneumoniae* resistant to ceftriaxone were isolated from one sample each.

The MAR indices for the five bacteria of significance to surveillance are given in Table 4. The study revealed that a significant number (57%) of all isolates had a MAR index greater than 0.2. *P. aeruginosa* bacterial isolates from SPF1 recorded the lowest MAR index of 0.1.

## 4. Discussion

This is the first study from Ghana to conduct surveillance of bacteria and WHO priority pathogens in effluents from seafood processing facilities. This study showed that effluents contained high bacterial loads, with *E. coli* being the predominant bacterial isolate present in almost all effluent samples and with high levels of antibiotic resistance. The study also confirmed the presence of WHO-priority pathogens in the effluent samples.

The major strength of this study is that effluent samples were collected from both seafood processing facilities according to international standards with quality control measures [13,19] under routine conditions, and therefore the results are valid and reliable in reflecting ground level reality. The antibiotics screened for resistance included a wide range of antibiotics belonging to all classes of the WHO AWaRe classification system [20], which are also commonly used in Ghana [20]. Therefore, the information on resistance patterns is highly relevant within the country context. The major limitation of this study is that the data were from a pilot surveillance project that collected effluent samples for 12 to 13 weeks in 2021 and 8 to 9 weeks in 2022. These weeks were not selected randomly and did not cover all seasons. Therefore, we are unable to comment on whether there are any seasonal variations in the presence or absence of bacteria such as *Shigella* spp. and *Salmonella* spp., which were not found in any of the effluent samples. Notwithstanding this limitation, the study findings have the following implications.

First, almost all of the effluent samples from both seafood processing facilities contained bacteria, with several samples showing a high bacterial load. This is similar to what has been found by studies in similar settings [6,10]. This indicates the presence of bacteria in the ecosystem of the seafood processing facilities. There is a need for further investigation into the source of these bacteria—whether the contamination is from the seafood animals that are undergoing processing, or from the water, or the equipment, or from the methods adopted by the personnel involved in the processing of the seafood. This will help in recommending and implementing appropriate measures to prevent contamination. Previous studies have shown that seafood processing equipment harbors several microbes, and the raw material used for preservation is the most common source of contamination [8,21].

Second, the high bacteria load also indicates that the treatment of effluents prior to discharge into the environment is ineffective in reducing the bacterial load. The consequences of the high load of bacteria in the local environment are unknown. If these effluents are discharged directly into the sea/ocean, it is not known whether these bacteria will survive and propagate in the sea/ocean environment. A study has shown that, due to the rich organic matter content of effluent from seafood processing facilities, the effluent has the ability to deplete dissolved oxygen levels and create an overall abundance of organisms in sea water [22]. At this stage, we recommend a review and rectification of the effluent treatment system to ensure reductions in the bacterial load of effluent.

Third, *E. coli*, *K. pneumoniae*, *Proteus* spp., *P. aeruginosa* and *A. baumannii*, known to adversely affect human health [23], were identified in the effluent samples. There is a need to assess if these bacteria are also present in the seafood produced from these facilities, which can lead to food contamination of the seafood products. Several studies in other countries have shown outbreaks of disease due to the consumption of contaminated seafood [24,25,26]. However, pathogens such as *Salmonella* spp., *Shigella* spp. and *Vibrio* spp. that are commonly associated with seafood contamination, as reported in one previous study [24], were not identified in any of the effluent samples.

Fourth, most of the bacteria were resistant to several antibiotics with a MAR index greater than 0.2. MAR index values greater than 0.2 indicate a high risk source of contamination where antibiotics are often used [27]. This indicates a need to investigate the reasons for such high levels of antibiotic resistance and whether the anecdotal information about non-usage of antibiotics in these two seafood processing facilities is indeed true. Multiple antibiotic resistance in bacteria is most commonly associated with the presence of plasmids that contain one or more resistance genes, each encoding for a single antibiotic resistance phenotype [27]. The injudicious use of antibiotics may propagate this antibiotic resistance within and between bacteria species. While there are no previous studies to show that antibiotics are used in seafood processing facilities, some studies have shown evidence of the use of antibiotics in fish farming [28].

Fifth, the most alarming finding from this study is the presence of WHO priority pathogens in a few effluent samples, most notably the *E. coli* resistant to cefotaxime, ceftazidime or carbapenem—the most potent currently available antibiotics in the treatment of infection by this bacterium. This calls for urgent measures to prevent the amplification/spread of these organisms from this local environment.

Sixth, our study, which uses the data from a pilot surveillance project, demonstrates the need for regularizing and expanding the scope of the surveillance system beyond effluents to cover all aspects of the seafood processing facilities and their products. Apart from enhancing the surveillance system for antibiotic resistant bacteria, our study also calls for establishing guidelines, providing state of the art equipment and building the capacity of the personnel associated with the seafood processing facilities to adopt best practices to enhance productivity and minimize unintentional bacterial contamination of the seafood products.

Lastly, although there were differences in the bacterial load and the bacterial species isolated between the two seafood processing facilities, we believe that these differences are not significant from the “One Health” point of view to warrant a differential approach in dealing with these two facilities. Furthermore, we believe that our attempt to use the operational research approach to describe, analyze and interpret the data from the pilot surveillance project has enhanced the scientific credibility of the results and the key recommendations from this pilot project. This approach may be expanded to all other spheres of the EPA’s activities in the country so that its decisions and activities are guided by scientific methods and evidence.

## 5. Conclusions

Data from the pilot surveillance project showed that the effluents of two seafood processing units contained a high bacterial load consisting of *Enterobacteriaceae* (*E. coli*., *Klebsiella* spp., *Proteus* spp.) more commonly, and *P. aeruginosa*, *A. baumannii* occasionally. These bacteria had high levels of multi-drug resistance with several of these species having a MAR index >0.2. The most significant finding was the presence of WHO priority pathogens—*E. coli* resistant to cefotaxime, ceftazidime or carbapenem and *K. pneumoniae* resistant to ceftriaxone—in some of the effluent samples. This calls for urgent measures, such as building capacity and strengthening surveillance across other industrial sectors to adopt best practices to prevent the augmentation and spread of antimicrobial resistance from these facilities.

## Figures and Tables

**Figure 1 ijerph-19-10823-f001:**
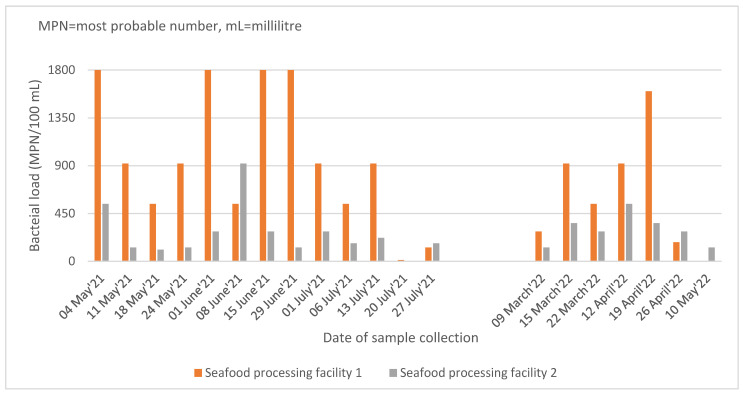
Bacterial load in the effluent samples collected from the two seafood processing facilities in Tema, Ghana during 2021 and 2022.

**Figure 2 ijerph-19-10823-f002:**
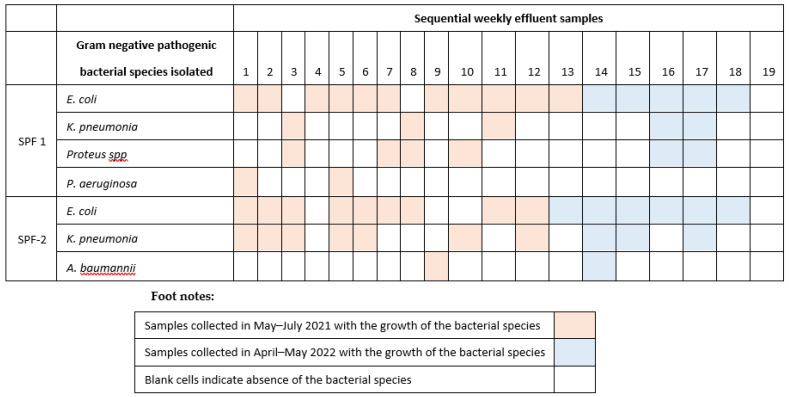
Gram-negative bacterial species of significance to human health isolated from the effluent samples collected from the two seafood processing facilities (SPF) in Tema, Ghana during 2021 and 2022.

**Table 1 ijerph-19-10823-t001:** Number of weekly effluent samples collected from the two seafood processing facilities (SPF) in Tema, Ghana during 2021 and 2022.

Time Period	Number of Weekly Effluent Samples Collected	
	SPF-1	SPF-2
May–July 2021	13	12
March–May 2022	6	7
Total	19	19

**Table 2 ijerph-19-10823-t002:** Bacterial species isolated from the effluent samples collected from the two seafood processing facilities (SPF) in Tema, Ghana during 2021 and 2022.

Bacterial Species Isolated	Number of Effluent Samples from Which the Bacterial Species Were Isolated	
	SPF-1 (*N* = 19)	SPF-2 (*N* = 19)
	*n* (%)	*n* (%)
*Escherichia coli*	16 (84)	15 (79)
*Klebsiella pneumoniae*	5 (26)	10 (53)
*Morganella morganii*	5 (26)	3 (16)
*Proteus mirabillis*	6 (32)	0
*Pseudomonas citronellolis*	6 (32)	0
*Citrobacter freudii*	4 (21)	2 (11)
*Pseudomonas putida*	1 (5)	3 (16)
*Enterobacter cloacae*	2 (11)	2 (11)
*Klebsiella oxytoca*	3 (16)	0
*Acinetobacter baumannii*	0	2 (11)
*Pseudomonas aeruginosa*	2 (11)	0
*Pseudomonas mendocina*	0	2 (11)
*Aeromonas jandaei*	0	2 (11)
*Klebsiella ozanae*	0	1 (5)
*Klebsiella aerogenes*	1 (5)	0
*Klebsiella variicola*	1 (5)	0
*Cronobacter* spp.	0	1 (5)

Footnotes: *N* = total number of effluent samples; *n* (%) = number (proportion) of effluent samples that contained the corresponding bacterial species out of the total effluent samples.

**Table 3 ijerph-19-10823-t003:** Antibiotic resistance patterns for Gram-negative bacteria species of significance to surveillance, which were isolated from effluent samples in two seafood processing facilities (SPF) in Tema, Ghana during 2021 and 2022.

Antibiotics	Isolates Resistant to Antibiotics						
	*E. coli*		*K. pneumoniae*		*P. mirabillis*	*P. aeruginosa*	*A. baumannii*
	SPF 1	SPF 2	SPF 1	SPF 2	SPF 1	SPF1	SPF2
	(*N* = 16)	(*N* = 15)	(*N* = 5)	(*N* = 10)	(*N* = 6)	(*N* = 2)	(*N* = 2)
	*n* (%)	*n* (%)	*n* (%)	*n* (%)	*n* (%)	*n* (%)	*n* (%)
Access							
Tetracycline (TET)	13 (81)	10 (67)	0 (0)	9 (60) *	3 (50)	0 (0)	0 (0)
Sulfamethoxazole/Trimethoprim (SXT)	9 (56)	8 (53)	1(20)	9 (60)	0 (0)	0 (0)	0 (0)
Gentamicin (GEN)	0 (0)	0 (0)	0 (0)	0 (0)	0 (0)	0 (0)	0 (0)
Ampicillin (AMP)	10 (63)	13 (87)	4(80)	10(67)	3(50)	0 (0)	2 (13)
Amikacin (AMK)	2 (13)	0 (0)	0 (0)	0 (0)	0 (0)	0 (0)	0 (0)
Watch							
Cefuroxime (CXM)	12 (75)	15(100)	4(80)	10(67)	2(33)	0 (0)	2 (13)
Cefotaxime (CTX)	6 (38)	2 (13)	0 (0)	0 (0)	0 (0)	0 (0)	0 (0)
Ciprofloxacin (CIP)	3 (19)	2 (13)	0 (0)	1 (7)	0 (0)	0 (0)	0 (0)
Levofloxacin (LEV)	5 (31)	5 (33)	0 (0)	0 (0)	0 (0)	0 (0)	0 (0)
Ceftazidime (CAZ)	0 (0)	1 (7)	0 (0)	0 (0)	0 (0)	0 (0)	0 (0)
Piperacillin/Tazobactam (TZP)	5 (31)	5 (33)	0 (0)	9 (60) *	0 (0)	2 (13)	0 (0)
Cefepime (FEP)	3 (19)	2 (13)	1(20)	0 (0)	0 (0)	0 (0)	0 (0)
Meropenem (MEM)	1 (6) *	0 (0)	0 (0)	0 (0)	0 (0)	0 (0)	0 (0)
Ceftriaxone (CRO)	0 (0)	0 (0)	1(20)	0 (0)	0 (0)	0 (0)	2 (13)
Reserve							
Colistin (CL)	0 (0)	0 (0)	0 (0)	0 (0)	0 (0)	0 (0)	0 (0)

* *p* < 0.05. Footnotes: N = total number of effluent samples; n (%) = number (proportion) of effluent samples that contained the corresponding bacterial species out of the total effluent samples.

**Table 4 ijerph-19-10823-t004:** Multiple antibiotic resistance indices (MAR) of the Gram-negative bacteria species of significance to human health isolated from effluent samples collected from two seafood processing facilities (SPF) in Tema, Ghana during 2021 and 2022.

Bacteria Isolates	Source	MAR Index
*E. coli*	SPF-1	0.7
	SPF-2	0.7
*K. pneumoniae*	SPF-1	0.3
	SPF-2	0.4
*Proteus* spp.	SPF-1	0.2
	SPF-2	No isolate
*A. baumannii*	SPF-1	No isolate
	SPF-2	0.2
*P. aeruginosa*	SPF-1	0.1
	SPF-2	No isolate

## Data Availability

The dataset used in this paper has been deposited at https://doi.org/10.6084/m9.figshare.20517912.v2 accessed on 19 August 2022 and is available under a CC BY 4.0 license.

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
