# Peer review of "Surveillance of WHO Priority Gram-Negative Pathogenic Bacteria in Effluents from Two Seafood Processing Facilities in Tema, Ghana, 2021 and 2022: A Descriptive Study"

_ijerph, 2022, doi:10.3390/ijerph191710823_

Round 1
Reviewer 1 Report
- Many typos in the text and figure
- Figure 1: the X axis should be “date of sample collection” and the position should not be too close to the legend showing the color code.
- In material and method or in table 3, authors should mention the concentration used for each antibiotic
- Inconsistency for the complete term of MAR, in material and method its written multi antibiotic resistance while in table 4 its written multidrug resistance
- The pH and temperature data also need to be provided as a supplementary file as authors mentioned they measured them in situ.
- Authors need to discuss why some bacteria are present in one SPF and missing in another one. In order to give a better insight to the readers, authors need to describe the difference between SPF-1 and SPF-2.
Reviewer 2 Report
ijerph-1873579. review
Title: Surveillance of WHO priority Gram negative….
Decision: Accept with minor revisions
The paper is well-written with sufficient and appropriate references. The conclusions are well-supported by the results. The paper addresses an important and critical health concern and is appropriate for the journal. A suggestion for minor improvements include:
1. In Section 2.3: Provide more precise detail on the sample collection location. Was is from the communal channel ? Was it prior to the treatment system currently in place ? Later discussion (e.g. lines 272 and 273) refers to such elements of the effluent discharge but the precise location in relation to these elements is currently unclear. Consider a diagram of the key systems and the sample location indicated on that diagram.
2. In Table 2 and 3. I suggest clearly defining the meaning of n%. Add this clarification into the table description/title.
3. Line 281 to 288 – Great observation
4. In future work it would be great to see some genetic analysis undertaken.
